# Discovery of Pathogenic Variants Associated with Idiopathic Recurrent Pregnancy Loss Using Whole-Exome Sequencing

**DOI:** 10.3390/ijms25105447

**Published:** 2024-05-17

**Authors:** Jeong Yong Lee, JaeWoo Moon, Hae-Jin Hu, Chang Soo Ryu, Eun Ju Ko, Eun Hee Ahn, Young Ran Kim, Ji Hyang Kim, Nam Keun Kim

**Affiliations:** 1Department of Biomedical Science, College of Life Science, CHA University, Seongnam 13488, Republic of Korea; smilee3625@naver.com (J.Y.L.); regis2040@nate.com (C.S.R.); ejko05@naver.com (E.J.K.); 2Endomics, Inc., Seongnam-si 13595, Republic of Korea; masicer@goendomics.com (J.M.); haejinhu@goendomics.com (H.-J.H.); 3Department of Obstetrics and Gynecology, CHA Bundang Medical Center, School of Medicine, CHA University, Seongnam 13596, Republic of Korea; bestob@cha.ac.kr (E.H.A.); happyiran@cha.ac.kr (Y.R.K.)

**Keywords:** bioinformatics, recurrent spontaneous abortion, miscarriage

## Abstract

Idiopathic recurrent pregnancy loss (RPL) is defined as at least two pregnancy losses before 20 weeks of gestation. Approximately 5% of pregnant couples experience idiopathic RPL, which is a heterogeneous disease with various causes including hormonal, chromosomal, and intrauterine abnormalities. Although how pregnancy loss occurs is still unknown, numerous biological factors are associated with the incidence of pregnancy loss, including genetic variants. Whole-exome sequencing (WES) was conducted on blood samples from 56 Korean patients with RPL and 40 healthy controls. The WES data were aligned by means of bioinformatic analysis, and the detected variants were annotated using machine learning tools to predict the pathogenicity of protein alterations. Each indicated variant was confirmed using Sanger sequencing. A replication study was also conducted in 112 patients and 114 controls. The Variant Effect Scoring Tool, Combined Annotation Dependent Depletion tool, Sorting Intolerant from Tolerant annotation tool, and various databases detected 10 potential variants previously associated with spontaneous abortion genes in patients by means of a bioinformatic analysis of WES data. Several variants were detected in more than one patient. Interestingly, several of the detected genes were functionally clustered, including some with a secretory function (mucin 4; *MUC4*; rs200737893 G>A and hyaluronan-binding protein 2; *HABP2*; rs542838125 G>T), in which growth arrest-specific 2 Like 2 (*GAS2L2*; rs140842796 C>T) and dynamin 2 (*DNM2;* rs763894364 G>A) are functionally associated with cell protrusion and the cytoskeleton. ATP Binding Cassette Subfamily C Member 6 (*ABCC6)* was the only gene with two variants. *HABP2* (rs542838125 G>T), *MUC4* (rs200737893 G>A), and *GAS2L2* (rs140842796 C>T) were detected in only the patient group in the replication study. The combination of WES and machine learning tools is a useful method to detect potential variants associated with RPL. Using bioinformatic tools, we found 10 potential variants in 9 genes. WES data from patients are needed to better understand the causes of RPL.

## 1. Introduction

Recurrent pregnancy loss (RPL) or recurrent spontaneous abortion (RSA) is defined as at least two consecutive failed pregnancies within 20 weeks of gestation [1,2]. Approximately 5% of pregnant couples experience RPL [1,3,4]. Several factors are associated with RPL, including chromosomal, uterine, and hormonal abnormalities, infection, immune responses, and environmental conditions. However, the etiology of the disease is unexplained in more than 50% of couples diagnosed with RPL.

Many recent studies have reported single-nucleotide polymorphisms associated with the pathogenicity of various diseases including RPL [5,6,7]. Candidate variants with known ontology have previously been characterized into molecular function, biological process, or cellular component categories. Currently, advanced technologies, such as next-generation sequencing (NGS) and chromosome microarray analysis, are used to discover potential disease-associated variants, including those for an RPL diagnosis [8,9]. Whole-exome sequencing is an NGS method that allows the screening of many variants and the discovery of potential novel human disease variants. Therefore, it is a useful tool to identify protein-encoding and associated pathogenic variants.

NGS technology has become a new paradigm for the diagnosis of genetic diseases [10]. Advanced NGS technologies include various methods developed to screen pathogenic variants according to the effects of amino acid changes. Although predicting causal variants is difficult, various methods exist utilizing computer programs or machine learning algorithms which typically predict structural results after sequence alteration or analyze known phenotype data after alterations. In this study, we used four tools to predict the effects of amino acid changes, including Combined Annotation Dependent Depletion (CADD) [11,12], Polymorphism Phenotyping v2 (Polyphen-2) [13], Sorting Intolerant from Tolerant (SIFT) [14], and the Variant Effect Scoring Tool (VEST) [15,16]. CADD is a tool used to score the deleteriousness of single-nucleotide and insertion/deletion (indel) variants in the human genome [11]. PolyPhen-2 is a tool that predicts the possible impact of an amino acid substitution on the structure and function of a human protein [13]. The SIFT algorithm uses sequence homology to predict how an amino acid substitution will affect protein function and potentially alter its phenotype [14]. VEST is a random forest machine learning algorithm used to identify functional missense mutations and classify them into grades [16].

In this study, we analyzed whole-exome data from 56 patients with RPL and 40 controls to screen potential rare variants associated with RPL. After annotating protein-coding variants, we filtered the variants using various bioinformatics tools and identified RPL-associated genes using open databases. According to the bioinformatic results, we conducted a replication study, using Genotyping by Taq-man assay and SNaPshot Multiplex assay, with a large sample size including 112 patients and 114 controls, and discovered three potential pathogenic variants found only in patients.

## 2. Results

NGS methods were used to identify potential pathogenic variants in samples from 56 RPL patients and 40 controls (Figure 1). All of the selected genes were rare variants (minor allele frequency < 0.01) with the cutoff criteria of CADD under 20, VEST p-value under 0.05, SIFT > 0.05, and a PolyPhen v2 of “possibly damaging”. After filtering, we confirmed the variant allele frequency to identify false positives with a cutoff value of 30%. We screened 10 variants and 9 genes, and all variants were heterozygous (Table 1). The filtered variants consisted of eight missense variants and two stop-gain variants; therefore, Chr3:195488449 and Chr 20:48130848 did not have SIFT and PolyPhen-2 scores. The Chr16:16276686, rs758166222 A>T variant was detected in two samples. Two variants in the *ABCC6* gene (Chr16:16276686 and Chr16:16282707) were detected in three samples. The *MUC4*, Chr3:195488449 variant had the highest CADD score (PHRED-like score = 51) and the lowest VEST p-value (0.006). However, because this was a stop-gain variant, it had no PolyPhen-2 or SIFT score. The Chr19:10897280 rs763894364 G>A variant had the lowest minor allele frequency (0.0005) in the Korea1K database.

We conducted a replication study of 10 rare variants previously associated with RPL, using the RPL and control samples (Table 2). As a result, only the patients of group 2 (rs200737894 G>A, rs542838125 G>T, rs140842796 C>T) showed continuous bioinformatics, but other variants were detected in the controls.

## 3. Discussion

In this study, we recruited 96 participants, including 56 RPL patients and 40 controls, and conducted whole-exome sequencing to screen potential causal variants for RPL. Ten variants were selected as potentially pathogenic variants that were previously reported with RPL-related genes. Among these genes, Chr16:16276686 (rs758166222 A>T; *ABCC6*; p.K682M) was the only variant found in two individuals, and the *ABCC6* gene was the only gene for which two variants were detected (Chr16:16276686 and Chr16;16282707) (Table 1). The *ABCC6* gene encodes ATP binding cassette subfamily C member 6, which is associated with transporter activity, and is a known causal gene of pseudoxanthoma elasticum [17,18]. Previous reports have described pseudoxanthoma elasticum as an autosomal recessive disorder that causes skin abnormalities. Additionally, it can occur after various pregnancy complications and result in miscarriage [18,19]. Among the nine detected genes, the variants found in our group were different from those reported previously in the same genes. Mucin 4 belongs to the *MUC* gene series (MUC1 to 22) and is associated with the endometrial epithelium. It has also been reported to be involved in recurrent implantation failure [20].

In our previous report, the rs882605 G>T and rs1104760A>G variants were associated with a subgroup of patients who experienced pregnancy loss more than twice [21]. Alkaline phosphatase, germ cell (*ALPG*), also known as Alkaline phosphatase, placental like-2 (*ALPPL2*), is highly involved in trophectoderm expression in the human placenta and is associated with early human development and the first cellular differentiation in the inner cell mass. A single-nucleotide polymorphism of this gene (rs11678251) is associated with placental reduction [22,23]. Prostaglandin I2 synthase (*PTGIS*) is involved in fetal development because prostaglandin synthesis is important in early pregnancy to establish implantation and placentation. Additionally, the overexpression of *PGE2* and other inflammatory cytokines in endometrial tissue results in a risk of spontaneous abortion [24]. Dynamin 2 (*DNM2*) regulates the invagination and constriction of vesicles at the membrane in endothelial and epithelial cells. Additionally, *DNM2* GTPase is ubiquitously expressed and regulates riboflavin absorption in the human placental trophoblast model cell line BeWo [25]. *CYP24A1* encodes vitamin D-24-hydroxylase, which is very important clinically and physiologically because it regulates vitamin D metabolism [26]. *CYP24A1* can be stimulated hormonally, and its mutation results in hypercalcemia in pregnancy [26]. In a previous study, the placental and decidual expression of *CYP24A1* was increased in patients with spontaneous miscarriage compared with that in healthy controls [27].

In the replication study, variants in the hyaluronan-binding protein 2 (*HABP2*), *COL6A3*, and *GAS2L2* genes were only detected in the patient group. *HABP2* and *COL6A3* are associated with cell adhesion, especially *HABP2*, which was recently reported to be associated with recurrent implantation failure [28]. Reduced *HABP2* expression in the endometrium results in a decrease in endometrium receptivity and in vitro fertilization failure [28,29]. *COL6A3* is a collagen VI-encoding gene that is an important component of the extracellular matrix and is usually found in interstitial tissue such as muscle, skin, and tendons [30]. An abnormally increased deposition of collagen disrupts uterine function, resulting in interrupted vascularization and delayed remodeling during early pregnancy [31]. The growth arrest-specific 2-like 2 (*GAS2L2*) protein is localized in actin stress fibers and microtubules, and the absence of *GAS2L2* in mice resulted in neonatal death [32]. It is also associated with a causal gene of primary ciliary dyskinesia, which is associated with miscarriage [33]. The rs758166222 A>T (*ABCC6*), rs763894364 G>A (*DNM2*), rs575918099 G>A (*ALPG*), and rs114476330 G>A (*CYP24A1*) variants were not detected in either patient or control groups. Thus, additional patients and controls are needed to examine these four variants.

We discovered possibly causative rare variants by means of whole-exome sequencing analysis in 56 non-related Korean RPL patients and 40 controls. In the process, we used bioinformatics tools and population data and these tools helped us find rare pathogenic variants. The prediction of disease risk using variants is difficult, but using whole-exome sequencing and bioinformatics tools is a good approach to finding pathogenic variants.

## 4. Materials and Methods

### 4.1. Subjects

The study group consisted of 168 women with at least two consecutive pregnancy losses before 20 weeks of gestation, who were diagnosed with idiopathic RPL according to the definition of the American Society for Reproductive Medicine, and 154 healthy controls. The patients were enrolled in the study at the Infertility Medical Center of CHA Bundang Medical Center from March 1999 to February 2010. Women in the control group were recruited from CHA Bundang Hospital and met the following criteria: history of at least one spontaneous pregnancy; current pregnancy; regular menstrual cycles; karyotype 46, XX; and no history of miscarriage. The study was approved by the Institutional Review Board of CHA Bundang Medical Center (IRB approval no. BD2010-123D), and written informed consent was obtained from all participants. Pregnancy loss was diagnosed based on the results of human chorionic gonadotropin tests, ultrasound, and/or physical examination before 20 weeks of gestation. None of the participants had a history of smoking or alcohol use. The following parameters were also measured using participant blood samples: activated partial thromboplastin time, body mass index, blood urea nitrogen, creatinine, estradiol, follicle-stimulating hormone, luteinizing hormone, platelet count, and prothrombin time.

Patients with the following conditions were excluded from the study: RPL or implantation failure due to hormonal, genetic, anatomic, infectious, autoimmune, or thrombotic causes. Anatomic causes were evaluated using hysterosalpingogram, hysteroscopy, computed tomography, and magnetic resonance imaging to detect intrauterine adhesions, septate uterus, and uterine fibroids. Hormonal causes, including hyperprolactinemia, luteal insufficiency, and thyroid disease, were evaluated using blood analyses. Infectious causes, such as the presence of *Ureaplasma urealyticum* or *Mycoplasma hominis*, were evaluated by means of bacterial culture. Autoimmune causes, including antiphospholipid syndrome or lupus, were evaluated using lupus anticoagulant and anticardiolipin antibodies. Thrombotic causes, such as thrombophilia, were evaluated through the identification of protein C and S deficiencies and the detection of β-2-glycoprotein 1 antibodies.

### 4.2. Whole-Exome Sequencing

Whole-exome sequencing was conducted on 56 patients and 40 controls. Genomic DNA was extracted from anticoagulated peripheral blood using a G-DEX blood extraction kit (Intron). A minimum of 500 ng of genomic DNA per sample was used to construct sequencing libraries. Whole-exome sequencing was conducted by Macrogen, Inc. SureSelect V4+UTR-post libraries were constructed according to the manufacturer’s recommendations (Agilent Technologies, Santa Clara, CA, USA) and were sequenced using a Novaseq system according to the manufacturer’s suggested protocol (Illumina Inc., SanDiego, CA, USA).

Trim_galore (version 0.6.4) was used to assess the quality of reads. The read sequences were aligned to the human reference genome (GRch37/hg19) using BWA (version 0.7.17-r1188). The sorting and indexing of BAM files were performed using SAMtools, and the BaseRecalibrator tool of GATK (version 4.3.0) and dbSNP (https://www.ncbi.nlm.nih.gov/snp/ (accessed on 21 September 2022) were used to annotate functionality, single-nucleotide variants, and deduplication. Variant calling after GATK analysis and functional annotation was performed using SnpEff tools (version 5.1).

### 4.3. Annotation and Variant Filtering

Before screening for predicted pathogenic single-nucleotide variants, only protein-coding gene variants, such as nonsynonymous, splicing variant, stop-gain, stop-loss, frameshift, and indel mutants, were selected. Variants were filtered according to a minor allele frequency > 0.01 in the 1000 Genomes database [34,35], and Korean-specific rare variants were selected using the Korean Reference Genome Database (KRGDB) [36] and Korea1K [37]. To select RPL-related genes, we collected data from various databases, including Disgenet, ClinVar, and Monarch, and excluded non-RPL genes (Appendix A).

### 4.4. Additional Annotation for Potential Variants

Pathogenic prediction annotation was used to select potential pathogenic variants. Common variants (minor allele frequency > 0.01) were filtered using population databases including the KRGDB and Korea1K databases (Table 1). Additionally, pathogenic annotation was conducted using the CADD, VEST, SIFT, and PolyPhen-2 tools to select pathogenic variants. The CADD tool integrates information contained in diverse annotations into a single score and prioritizes functional and pathogenic variants [38]. Variants with a CADD PHRED-like score ≥ 20 are classified as pathogenic [11]. VEST is a computational tool for the identification of specific variants that contribute to human disease [16]. It characterizes variants into categories (missense, frameshift, indel, stop-gain, stop-loss, and splice site), and the pathogenicity of genes is integrated into a *p*-value (non-silent). Variants with scores ≥ 0.5 in each category and an integrated *p*-value of ≤0.05 are classified as pathogenic. These tools are machine learning-based and suggest a score for each variant, including nonsynonymous, indel, and splice variants, regardless of whether they are deleterious or pathogenic. The SIFT algorithm predicts the potential effects of amino acid changes on protein function (http://sift-dna.org (accessed on 31 March 2022)). It characterizes missense and indel variants using a score ranging from 0.0 to 1.0. Variants with a score close to 0.00 are deleterious, but variants with a score ≥ 0.05 are predicted to be tolerated (benign) [14,39]. The PolyPhen-2 tool predicts the effects of amino acid substitutions on protein structure and function (http://genetics.bwh.harvard.edu/pph2/ (accessed on 21 June 2021). The score ranges from 0.0 to 1.0 which are classified into three categories: benign < 0.15; possibly damaging = 0.15–1.0; and probably damaging > 0.85 [13]. Function-related gene analysis was performed using the Database for Annotation, Visualization and Integrated Discovery (DAVID) [40,41], and string interaction network analysis [42].

### 4.5. Validation

Variants identified using whole-exome sequencing that were found in both controls and patients were excluded. Before capillary sequencing, we calculated variant allele frequency and excluded false positives with a cutoff of 30%. We validated candidate variants by means of capillary sequencing, using an Applied Biosystems 3730xl DNA analyzer. All primers were designed using Primer3Plus web software (https://www.primer3plus.com/primer3plusAbout.html (ver 3.3.0) [43]. Each primer sequence is shown in Appendix A. Data obtained by means of capillary sequencing were analyzed using Chromas (version 2.6.6 Technelysium Pty Ltd., Brisbane, Australia).

### 4.6. Replication of Patient and Control Samples

After the annotation and filtering of whole-exome sequencing data, to see if this result could also be verified in a larger sample size, a replication study was conducted using RPL and control samples (112 patients and 114 controls). Genotyping was performed using the SNaPshot multiplex system for rs140842796 C>T, rs13306027 G>T, and rs114476330 G>A; a Taqman assay for rs575918099 G>A, rs200737893 G>A, rs542838125 G>T, rs758166222 A>T, rs527236047 C>G, and rs763894364 G>A; and PCR-restriction fragment length polymorphism for rs116238578 G>A.

## 5. Conclusions

This is the first report of a case–control study utilizing bioinformatic tools. Most studies, especially case–control studies, target specific already known genes, SNPs, or pathways; however, we used whole-exome sequencing data and discovered novel potential pathogenic variants. These results provide a strategy for discovering novel pathogenic variants. Idiopathic RPL is a heterogeneous disorder that is affected by various factors. We conducted whole-exome sequencing of 56 patients and discovered 10 variants according to population frequency, using various annotation tools including CADD, VEST, SIFT, and PolyPhen-2, that were previously reported to be associated with RPL. In a replication study, only the *HABP2*, *GAS2L2*, and *MUC4* variants were detected in the patient group. To increase the accuracy of our analysis, additional population information should be collected and validated using databases, and the study should be replicated with additional patients and controls.

## Figures and Tables

**Figure 1 ijms-25-05447-f001:**
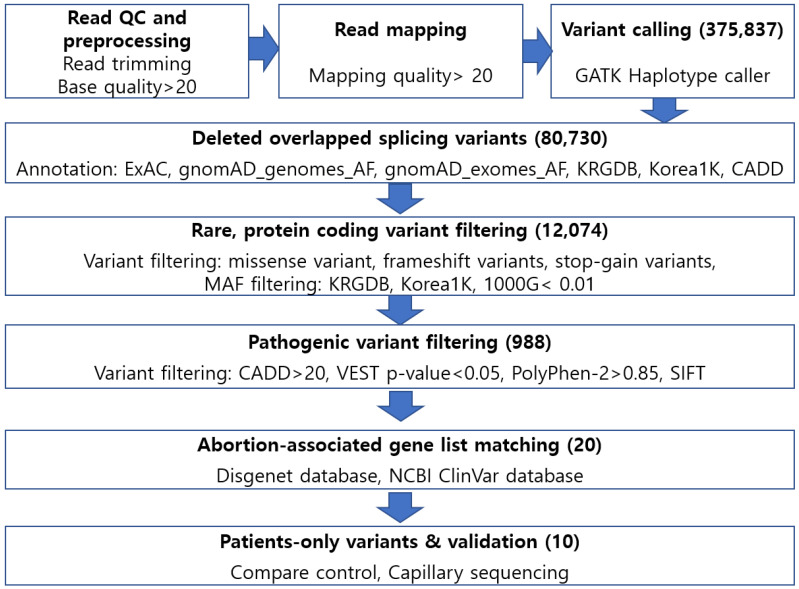
Whole-exome sequencing analysis pipeline. An abortion-associated, matching gene list was filtered using the Disgenet and ClinVar databases and the results were compared with those in the control sample to select patient-specific variants. Combined Annotation Dependent Depletion: CADD; Variant Effect Scoring Tool: VEST; Sorting Intolerant from Tolerant: SIFT.

**Table 1 ijms-25-05447-t001:** List of potential recurrent pregnancy loss candidate genes and information in 56 patients and 40 controls.

Position	rsID	Gene	Transcript_ID	HGVS.c	HGVS.p		MAF frequency				Pathogenicity						
						Replication Study	1000 Genomes	KRGDB	KOREA1K	gnomAD Genomes_AF	VEST *p*-Value	VEST Missense	VEST Stop-Gain	CADD	SIFT	Polyphen2	Detected Individual Count
chr2 233274367	rs575918099	ALPG	NM_031313.3	c.1384G>A	p.G462S	.	0.0002	0.0038	0.0027	.	0.0170	0.775	.	24.4	0	1	1
chr2 238283646	rs116238578	COL6A3	NM_057165.5	c.2470G>A	p.V824M	0.01327	0.0016	0.0013	0.0087	0.0003	0.0207	0.737	.	23.8	0	0.995	1
chr3 195488449	rs200737893	MUC4	NM_138297.5	c.1659G>A	p.W553 *	0.00443	0.0004	0.0013	0.0027	0.0000	0.0060	.	0.634	51	N/A	N/A	1
chr10 115334164	rs542838125	HABP2	NM_001177660.2	c.145G>T	p.D49Y	0.00443	.	0.0013	0.0005	0.0000	0.0150	0.799	.	32	0	0.987	1
chr16 16276686	rs758166222	ABCC6	NM_001171.5	c.2045A>T	p.K682M	.	.	0.0010	0.0011	.	0.0145	0.805	.	24.5	0	1	2
chr16 16282707	rs527236047	ABCC6	NM_001171.5	c.1760C>G	p.S587C	0.01327	.	0.0025	0.0022	0.0000	0.0398	0.613	.	24	0.03	0.979	1
chr17 34074883	rs140842796	GAS2L2	NM_139285.4	c.817C>T	p.R273C	0.00885	0.0018	0.0013	0.0044	0.0004	0.0075	0.93	.	28.7	0	1	1
chr19 10897280	rs763894364	DNM2	NM_001190716.2	c.890G>A	p.R297H	.	0.0002	0.0013	0.0005	0.0000	0.0129	0.827	.	28.6	0	0.942	1
chr20 48130848	rs13306027	PTGIS	NM_000961.4	c.940G>T	p.E314 *	0.00443	.	0.0013	0.0011	0.0000	0.0177	.	0.569	33	N/A	N/A	1
chr20 52789538	rs114476330	CYP24A1	NM_001128915.2	c.359G>A	p.R120H	.	0.0002	0.0013	0.0033	0.0000	0.0159	0.788	.	32	0	1	1

VEST analysis generates values between 0 and 1, and VEST *p*-values ≥ 0.5 were classified as pathogenic. Scores ≥ 0.5 were classified as pathogenic and those < 0.5 as benign. CADD analysis generates a PHRED-like scaled value. Scores ≥ 20 were classified as pathogenic variants and those < 20 as benign. SIFT analysis generates values between 0 and 1, and scores ≥ 0.5 were classified as pathogenic. Polyphen2 analysis generates value between 0 and 1, and scores are matched with the RPL-associated gene list. * is stop-gain alteration that indicate ‘stop’ after alteration of nucleotide.

**Table 2 ijms-25-05447-t002:** Replication of rare variants in 112 patients and 114 controls.

Groups	Gene (rsID) (# of Patients/# of Controls)			
Group 1	ABCC6 (rs758166222)	DNM2 (rs763894364)	ALPG (rs575918099)	CYP24A1 (rs114476330)
	0/0	0/0	0/0	0/0
Group 2	MUC4 (rs200737893)	HABP2 (rs542838125)	GAS2L2 (rs140842796)	
	1/0	1/0	2/0	
Group 3	ABCC6 (rs527236047)	PTGIS (rs13306027)		
	0/3	0/1		
Group 4	COL6A3 (rs116238578)			
	2/1			

Group 1: Not detected in both controls and patients. Group 2: Detected only in patients and not in controls. Group 3: Detected only in controls and not in patients. Group 4: Detected in both controls and patients.

## Data Availability

The datasets generated and/or analyzed during the current study are not publicly available due to another publication and personal information, but are available from the corresponding author on reasonable request.

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
