# Peer review of "Discovery of Pathogenic Variants Associated with Idiopathic Recurrent Pregnancy Loss Using Whole-Exome Sequencing"

_ijms, 2024, doi:10.3390/ijms25105447_

Round 1

Reviewer 1 Report (Previous Reviewer 3)

Comments and Suggestions for Authors

Although some of the efforts can be seen from the authors to improve the manuscript, I still cannot find a data availability term throughout the manuscript for the WES data. If the study includes only the dataset from public database, it will hugely dampen the novelty of the study.

Author Response

Reviewer 1.

Although some of the efforts can be seen from the authors to improve the manuscript, I still cannot find a data availability term throughout the manuscript for the WES data. If the study includes only the dataset from public database, it will hugely dampen the novelty of the study.

As we mentioned all our data is about Korean RPL patients and the whole exome sequencing data also about 56 Korean RPL patients which is visit our medical center from 1999 to 2010.

We cannot open data as public because we are still analyzing and planning to submit more But available with reasonable request, to corresponding author.

We are notice Data availability section as below

The datasets generated and/or analysed during the current study are not publicly available due to another publication and personal information but are available from the corresponding author on reasonable request.

Reviewer 2 Report (Previous Reviewer 2)

Comments and Suggestions for Authors

This is a resubmission of a manuscript. Over the review process, the manuscript has gone through extensive changes. There are minor issues to be noted mainly. The authors must point out the novelty of the field and why there is a need for communication. 

Abstract: OK

Introduction: line 97 -99 this phrase should be rerwitten

Methods: OK

Results: OK

Discussions: In this part or the introductory part should be stressed out the novelty of the findings and, therefore, the need for communication

Conclusion: OK

Author Response

Reviewer 2.

This is a resubmission of a manuscript. Over the review process, the manuscript has gone through extensive changes. There are minor issues to be noted mainly. The authors must point out the novelty of the field and why there is a need for communication. 

Abstract: OK

Introduction: line 97 -99 this phrase should be rerwitten

Thank you for your comments we are add more information about replication study in Line 71

Genotyping by Taq-man assay and SNaPshot assay with a large sample size including 112 patients and 114 controls and discovered three potential pathogenic variants found only in patients.

Methods: OK

Results: OK

Discussions: In this part or the introductory part should be stressed out the novelty of the findings and, therefore, the need for communication

Thank you for your comment and we add paragraph end of discussion that point of our study

We are discovered a possibly causative rare variants by whole exome sequencing analysis in not related 56 RPL patients and 40 controls Korean. In the progress, we are using bioinformatics tools and population data and these tools help to find pathogenic rare variants. Prediction of risk in disease by variants is difficult but using a whole exome sequencing and with bioinformatics tools are good approach to find pathogenic variants.

Conclusion: OK

This manuscript is a resubmission of an earlier submission. The following is a list of the peer review reports and author responses from that submission.

Round 1

Reviewer 1 Report

Comments and Suggestions for Authors

The authors discuss their study using whole exome sequencing to establish whether there are particular genes that are implicated in recurrent pregnancy loss.  They have used a variety of software tools to predict variant genes and have looked at a second set of samples from patients and controls for validation.

Comments:

Tables 1 and 2 have now been included in the manuscript but there remain some inconsistencies.

Lines 45/46 – please make it clearer that you are talking about RPL here.

Paragraph starting on line 50 – I would suggest discussing bioinformatics tools/approaches, as opposed to NGS technology here.

Lines 65-68 – mention replication study here too?

Figure 1 is not referred to from the text

Line 75 – please clarify whether this was a retrospective study given collection of samples between March 1999 and February 2010.  Please indicate the nature of the samples collected (peripheral blood?), how the samples were stored and how long the samples were kept in storage for before they were used.

Line 83 – please rephrase for clarity

Line 92 – please clarify why hg38 version of the human genome reference sequence was not used.

Line 100 – please add minor allele frequency information from dbSNP as well (https://www.ncbi.nlm.nih.gov/snp/)

Table 1 is now present but the table legend seems incomplete and has in some places repetitive information.  Citations should be included for the databases used.

Line 128 – ‘those with a score close to 1.0’ – please be more specific, for example, 0.85-1.0?

Lines 136/137 – please refer in text to Supplementary Table 2 for primer sequences

Section 2.6 – please clarify why different approaches for genotyping samples were used

Please give nucleotide and amino acid changes for rs numbers within the text.

Lines 165-167 – please rephrase for clarity

Table 2 – please include in table legend that where you have listed numbers separated by /, that the first number is number of patients and the second number is number of controls.

Other relevant studies to consider: https://pubmed.ncbi.nlm.nih.gov/34925444/

Comments on the Quality of English Language

There are a few sections of the manuscript as identified in comments that require attention to improve clarity.

Author Response

Reviewer 1.

The authors discuss their study using whole exome sequencing to establish whether there are particular genes that are implicated in recurrent pregnancy loss.  They have used a variety of software tools to predict variant genes and have looked at a second set of samples from patients and controls for validation.

Comments:

Tables 1 and 2 have now been included in the manuscript but there remain some inconsistencies.

Lines 45/46 – please make it clearer that you are talking about RPL here.

Thank you for your comments. I revised sentence and add reference line 44

Currently, advanced technologies, such as next-generation sequencing (NGS) and chromosome microarray analysis, are used to discover potential disease-associated variants, including those for an RPL diagnosis.

Paragraph starting on line 50 – I would suggest discussing bioinformatics tools/approaches, as opposed to NGS technology here.

Thank you for your comment, This paragraph is about introducing bioinformatics tools that I used in this paper, we added more information of bioinformatics tool as below line 50

Although predicting causal variants is difficult, various methods exist utilizing computer programs or machine learning which typically predict structural results after sequence alteration or analyze known phenotype data after alterations.

Lines 65-68 – mention replication study here too?

Thank you for your comment. I added mention about our replication study. Line 67

According to the bioinformatic results, we conducted a replication study with a large sample size and discovered three potential pathogenic variants.

Figure 1 is not referred to from the text

Thank you for your comment. we added Figure 1 in line 151

Line 75 – please clarify whether this was a retrospective study given collection of samples between March 1999 and February 2010.  Please indicate the nature of the samples collected (peripheral blood?), how the samples were stored and how long the samples were kept in storage for before they were used.

We collected blood samples and extract DNA within at least 2 days. Because of quality of DNA sample And as you know our data is about NGS data, our NGS data was built in 2012.

Line 83 – please rephrase for clarity

Thank you for your comment. I revised methods more clarify, line 81

Women in the control group were recruited from CHA Bundang Hospital and met the following criteria: history of at least one spontaneous pregnancy; current pregnancy; regular menstrual cycles; karyotype 46, XX; and no history of miscarriage. The study was approved by the Institutional Review Board of CHA Bundang Medical Center (IRB approval no. BD2010-123D), and written informed consent was obtained from all participants. Pregnancy loss was diagnosed based on the results of human chorionic gonadotropin tests, ultrasound, and/or physical examination before 20 weeks of gestation. None of the participants had a history of smoking or alcohol use. The following parameters were also measured using participant blood samples: activated partial thromboplastin time, body mass index, blood urea nitrogen, creatinine, estradiol, follicle-stimulating hormone, luteinizing hormone, platelet count, and prothrombin time.

Patients with the following conditions were excluded from the study: RPL or implantation failure due to hormonal, genetic, anatomic, infectious, autoimmune, or thrombotic causes. Anatomic causes were evaluated using hysterosalpingogram, hysteroscopy, computed tomography, and magnetic resonance imaging to detect intrauterine adhesions, septate uterus, and uterine fibroids. Hormonal causes, including hyperprolactinemia, luteal insufficiency, and thyroid disease, were evaluated by blood analyses. Infectious causes, such as the presence of Ureaplasma urealyticum or Mycoplasma hominis, were evaluated by bacterial culture. Autoimmune causes, including antiphospholipid syndrome or lupus, were evaluated using lupus anticoagulant and anticardiolipin antibodies. Thrombotic causes, such as thrombophilia, were evaluated by identification of protein C and S deficiencies and by detection of β-2-glycoprotein 1 antibodies.

Line 92 – please clarify why hg38 version of the human genome reference sequence was not used.

When use bioinformatic tools, we are confusing with hg38 version and more convenience when annotate population data especially Korean reference genome database (KRGDB) is built in GRCh37.

Line 100 – please add minor allele frequency information from dbSNP as well (https://www.ncbi.nlm.nih.gov/snp/)

Thank you for your comment, we added more information as below line 123

dbSNP  (https://www.ncbi.nlm.nih.gov/snp/)

Table 1 is now present but the table legend seems incomplete and has in some places repetitive information.  Citations should be included for the databases used.

Line 128 – ‘those with a score close to 1.0’ – please be more specific, for example, 0.85-1.0?

Thank you for your comment, we revise as below line 156

Variants with a score <0.15 are classified as benign, those with a score of 0.15 to 1.0 are classified as possibly damaging, and those with a score above 0.85 are considered probably damaging.

Lines 136/137 – please refer in text to Supplementary Table 2 for primer sequences

Thank you for your comment, we added information about primer in validation.

Section 2.6 – please clarify why different approaches for genotyping samples were used

Thank you for your comment. We want to find if these result could reached continuously in large sample and we added purpose in result line 169 as below

After annotation and filtering of whole-exome sequencing data, to see if this result could also be verified in a larger sample size, a replication study was conducted using RPL and control samples

Please give nucleotide and amino acid changes for rs numbers within the text.

We revise rs number with amino acid in text

Lines 165-167 – please rephrase for clarity

Thank you for your comment, we revise manuscript easily.

We conducted a replication study of 10 rare variants previously associated with RPL using the RPL and control samples (Table 2). As a result, only the patients of group 2 (rs200737894 G>A, rs542838125 G>T, rs140842796 C>T) showed continuous bioinformatics, but other variants were detected in the controls.

Table 2 – please include in table legend that where you have listed numbers separated by /, that the first number is number of patients and the second number is number of controls.

Thank you for your opinion. We added in table legend.

Number in bracket is number of patients and control (# of patients/# of controls)

Other relevant studies to consider: https://pubmed.ncbi.nlm.nih.gov/34925444/

Reviewer 2 Report

Comments and Suggestions for Authors

The manuscript is well-written. However, there are some issues before the manuscript can be published.

Abstract: OK

Introduction: OK

Methods:  The inclusion criteria should be presented more clearly

                    A flow chart will give a quick and good understanding of the study design.

                    Linwe 103-104  Were all genes colected from various databases tasted?

                    The validation process ( line 134) must be expended  - in a few phrases

Results: OK

Discussion: Communications usually present groundbreaking preliminary results or significant findings that are part of a more extensive study over multiple years. This aspect must be taken into account. And should be inserted in the manuscript.( what are the groundbreaking findings….)

Conclusions: Must be rewritten to reflect the topic of a Communication

Author Response

Reviewer 2

The manuscript is well-written. However, there are some issues before the manuscript can be published.

Abstract: OK

Introduction: OK

Methods:  The inclusion criteria should be presented more clearly

                    A flow chart will give a quick and good understanding of the study design.

                    Linwe 103-104 Were all genes colected from various databases tasted?

                    The validation process ( line 134) must be expended  - in a few phrases

Thank you for your comment, We add how to validate filtered variants as below in line 164

Before capillary sequencing, we calculated variant allele frequency with a cutoff of 35% and excluded variants after matching the whole-exome sequencing and NGS results.

Results: OK

Discussion: Communications usually present groundbreaking preliminary results or significant findings that are part of a more extensive study over multiple years. This aspect must be taken into account. And should be inserted in the manuscript.( what are the groundbreaking findings….)

When find pathogenic variants in case control study, usually seeking already known gene or already known mechanism or pathway. But Our data is about result of rare variants which is about RPL potentially pathogenic variants. And method to find a novel rare variants is not usually used.

Conclusions: Must be rewritten to reflect the topic of a Communication

Thank you for your comment and we add as below line 252

This is the first report of a case–control study utilizing bioinformatic tools. Most studies, especially case–control studies, target specific already known genes, SNPs, or signaling pathways; however, we used whole-exome sequencing data and discovered novel potential pathogenic variants. These results provide a strategy for discovering novel pathogenic variants.

Reviewer 3 Report

Comments and Suggestions for Authors

The study reported whole-exome sequencing data on 56 Korean RPL patients and 40 matched controls, and identified 10 potential variants in 9 genes associated with spontaneous abortion. The identified variants were further validated by capillary sequencing. The study provides the community with resources to a data of new cohort. However, there are a few points need to be improved regarding its nature of resources.

1. Given the nature that the study mainly includes the genomic data, the authors should included the links to access the data and a detailed introduction of how to download or access the WES data. Otherwise, the value of the study will be hugely dampened.

2. Though the authors described a workflow of variant call, however, no additional illustrations have been provided to evaluate the sequencing quality as well as the quality controls of the variant calls.

3. The authors annotated the identified variants from the study by a few public database, e.g. CADD, VEST. However, the authors did not make any clear conclusions from their results though they mentioned a larger population might be needed. Have these reported variants or their associated genes been described elsewhere in RPL studies?

Author Response

Reviewer 3.

The study reported whole-exome sequencing data on 56 Korean RPL patients and 40 matched controls, and identified 10 potential variants in 9 genes associated with spontaneous abortion. The identified variants were further validated by capillary sequencing. The study provides the community with resources to a data of new cohort. However, there are a few points need to be improved regarding its nature of resources.

  1. Given the nature that the study mainly includes the genomic data, the authors should included the links to access the data and a detailed introduction of how to download or access the WES data. Otherwise, the value of the study will be hugely dampened.

Thank you for your comment. we added information of each WES data easily access or added citation of each bioinformatics tools in reference

PolyPhen v2 - (http://genetics.bwh.harvard.edu/pph2/)

SIFT – (http://sift-dna.org)

  1. Ng, P.C.; Henikoff, S. SIFT: predicting amino acid changes that affect protein function. Nucleic Acids Res. 2003, 31, 3812–3814, doi:10.1093/nar/gkg509.

CADD-

  1. Rentzsch, P.; Schubach, M.; Shendure, J.; Kircher, M. CADD-Splice—improving genome-wide variant effect prediction using deep learning-derived splice scores. Genome Med. 2021, 13, 31, doi:10.1186/s13073-021-00835-9.
  2. Schubach, M.; Maass, T.; Nazaretyan, L.; Röner, S.; Kircher, M. CADD v1.7: using protein language models, regulatory CNNs and other nucleotide-level scores to improve genome-wide variant predictions. Nucleic Acids Res. 2024, 52, D1143–D1154, doi:10.1093/nar/gkad989.

VEST-

  1. Douville, C.; Masica, D.L.; Stenson, P.D.; Cooper, D.N.; Gygax, D.M.; Kim, R.; Ryan, M.; Karchin, R. Assessing the Pathogenicity of Insertion and Deletion Variants with the Variant Effect Scoring Tool (VEST-Indel). Hum. Mutat. 2016, 37, 28–35, doi:https://doi.org/10.1002/humu.22911.

  1. Though the authors described a workflow of variant call, however, no additional illustrations have been provided to evaluate the sequencing quality as well as the quality controls of the variant calls.

Thank you for your comment, We add how to validate filtered variants as below in line 164

Before capillary sequencing, we calculated variant allele frequency with a cutoff of 35% and excluded variants after matching the whole-exome sequencing and NGS results.

  1. The authors annotated the identified variants from the study by a few public database, e.g. CADD, VEST. However, the authors did not make any clear conclusions from their results though they mentioned a larger population might be needed. Have these reported variants or their associated genes been described elsewhere in RPL studies?

Thank you for your comment and we add as below line 252

This is the first report of a case–control study utilizing bioinformatic tools. Most studies, especially case–control studies, target specific already known genes, SNPs, or signaling pathways; however, we used whole-exome sequencing data and discovered novel potential pathogenic variants. These results provide a strategy for discovering novel pathogenic variants.

Reviewer 4 Report

Comments and Suggestions for Authors

Firstly, I would like to congratulate the authors of the manuscript on their very interesting study on

“Discovery of pathogenic variants associated with idiopathic recurrent pregnancy loss by whole-exome sequencing”. There are few minor observations that the authors should address before final submission.

1.            Avoid use of abbreviations in abstract.

2.            Avoid abbreviations at the start of a sentence.

3.            Line 26-27 rephrase sentence.

4.            Define abbreviations at first citation, throughout the manuscript and avoid abbreviations at the start of a sentence.

5.            Number of patients are not consistent. L 65-68 have a different patient number than the L 76 to 78?

6.            L 88 Avoid numeric at the start of a sentence, check throughout.

7.            Please add a valid statistical method used to analyze data?

8.            There is a need to improve results and discussion section as well.

9.            Authors should get help from academic writer for academic language editing before final submission, please.

Comments on the Quality of English Language

Minor academic English editing needed. 

Author Response

Firstly, I would like to congratulate the authors of the manuscript on their very interesting study on

“Discovery of pathogenic variants associated with idiopathic recurrent pregnancy loss by whole-exome sequencing”. There are few minor observations that the authors should address before final submission.

  1. Avoid use of abbreviations in abstract.

Thank you for your comment. we added full name of gene name.

  1. Avoid abbreviations at the start of a sentence.

Thank you for your comment we revise abbreviations when the word mentioned first in paper

  1. Line 26-27 rephrase sentence.

Thank you for your comment we repharase abstract

  1. Define abbreviations at first citation, throughout the manuscript and avoid abbreviations at the start of a sentence.

Thank you for your comment we revise abbreviations when the word mentioned first in paper

  1. Number of patients are not consistent. L 65-68 have a different patient number than the L 76 to 78?

Thank you for your comment. Our data consist bioinformatics of whole exome sequencing and replication study of validation within large sample. 168 patient samples consist of 56 NGS sample and 112 replication patients. 154 control samples consist 40 NGS samples and 114 replication controls

  1. L 88 Avoid numeric at the start of a sentence, check throughout.

Thank you for your comment, We change the sentence not start with number

  1. Please add a valid statistical method used to analyze data?

Thank you for your comment, We add how to validate filtered variants as below in line 135

Before capillary sequencing, we calculated variant allele frequency with a cutoff of 35% and excluded variants after matching the whole-exome sequencing and NGS results.

  1. There is a need to improve results and discussion section as well.

Thank you for your comment. we improve results.

  1. Authors should get help from academic writer for academic language editing before final submission, please.

Thank you for your comments we will get help to editing service soon.
